

# Antifungal properties of volatile organic compounds produced by *Daldinia eschscholtzii* MFLUCC 19-0493 isolated from *Barleria prionitis* leaves against *Colletotrichum acutatum* and its post-harvest infections on strawberry fruits

Sarunpron Khruengsai[1], Patcharee Pripdeevech[1,2], Chutima Tanapichatsakul[1], Chanin Srisuwannapa[1], Priya Esilda D'Souza[3] and Parinya Panuwet[3]

[1] School of Science, Mae Fah Luang University, Muang, Chiang Rai, Thailand
[2] Center of Chemical Innovation for Sustainability (CIS), Mae Fah Luang University, Muang, Chiang Rai, Thailand
[3] Laboratory of Exposure Assessment and Development for Environmental Research, Gangarosa Department of Environmental Health, Rollins School of Public Health, Emory University, Atlanta, Georgia, United States of America

Corresponding authors
Patcharee Pripdeevech,
patcharee.pri@mfu.ac.th
Parinya Panuwet,
ppanuwe@emory.edu,
parinya.panuwet@emory.edu

## ABSTRACT

Fungal endophytes are microorganisms living symbiotically with a host plant. They can produce volatile organic compounds (VOCs) that have antimicrobial activity. This study aimed to isolate endophytic fungi from *Barleria prionitis* plants grown in Thailand and to investigate the antifungal properties of their VOCs against *Colletotrichum acutatum*, a causal agent of anthracnose disease on post-harvest strawberry fruits. A total of 34 endophytic fungi were isolated from leaves of *B. prionitis*. The VOCs produced from each individual isolate were screened for their antifungal activity against *C. acutatum* using a dual-culture plate method. From this in vitro screening experiment, the VOCs produced by the endophytic isolate BP11 were found to have the highest inhibition percentage (80.3%) against the mycelial growth of *C. acutatum*. The endophytic isolate BP11 was molecularly identified as *Daldinia eschscholtzii* MFLUCC 19-0493. This strain was then selected for an in vivo experiment. Results from the in vivo experiment indicated that the VOCs produced by *D. eschscholtzii* MFLUCC 19-0493 were able to inhibit infections by *C. acutatum* on organic fresh strawberry fruits with an average inhibition percentage of 72.4%. The quality of the pathogen-inoculated strawberry fruits treated with VOCs produced by *D. eschscholtzii* MFLUCC 19-0493 was evaluated. Their fruit firmness, total soluble solids, and pH were found to be similar to the untreated strawberry fruits. Solid phase microextraction-gas chromatographic-mass spectrometric analysis of the VOCs produced by *D. eschscholtzii* MFLUCC 19-0493 led to the detection and identification of 60 compounds. The major compounds were elemicin (23.8%), benzaldehyde dimethyl acetal (8.5%), ethyl sorbate (6.8%), methyl geranate (6.5%), trans-sabinene hydrate (5.4%), and 3,5-dimethyl-4-heptanone (5.1%). Each major compound was tested for its antifungal activity against *C. acutatum* using the in vitro assay. While all these selected VOCs showed varying degrees of antifungal activity, elemicin was found to possess the strongest antifungal activity. This

work suggests that *D. eschscholtzii* MFLUCC 19-0493 could be a promising natural preservative for controlling *C. acutatum* associated anthracnose disease in strawberry fruits during the post-harvest period.

## INTRODUCTION

Strawberry (*Fragaria* × *ananassa* Duch.) is a common edible fruit of great economic importance. It can be cultivated throughout the world, but mostly in areas with temperatures ranging from 15 to 25 °C (*Natsheh, Abu Khalaf & Mousa, 2014*). Strawberries are a good source of vitamin C, folate, and phenolic compounds. Antioxidant compounds, such as anthocyanin and ellagitannins, are also detected in strawberry fruits in high amounts (*Giampieri et al., 2012*). In Thailand, strawberries can only be cultivated in the few northern and northeastern provinces that are at the appropriate elevations and have suitable ambient temperatures, such as Chiang Mai, Chiang Rai, and Loei (*Riyaphan, Pipattanawong & Subhadrabandu, 2004*). Due to high demand for domestic consumption, strawberries have been imported from several countries to Thailand with a total market value of hundreds of millions of Thai baht per year (*Pipattanawong, 2015*).

The quality of strawberries is determined by their firmness, taste, and flavor, as well as their color, shape, and the presence or absence of defects (*Kader, 1991*). However, infections by pathogens during the cultivation and post-harvest periods can drastically affect this quality (*Sistrunk & Morris, 1983*). The pathogen that usually causes significant fruit rot disease (anthracnose disease) in strawberries during the post-harvest period is the fungal pathogen *Colletotrichum acutatum*. Infection by this pathogen contributes to significant loss in the market value of fresh strawberries (*Arroyo et al., 2007*). Anthracnose disease also occurs in other fruits such as peaches, almonds, avocados, mangoes, papayas, and guavas, especially during the post-harvest period (*Regnier et al., 2010*). Chemical fungicides (e.g., azoxystrobin, pyraclostrobin, and captan) are typically used to control anthracnose disease in strawberries during cultivation (*Chalfoun, Castagnaro & Díaz Ricci, 2011*). For post-harvest prevention of anthracnose disease, methyl bromide has been used. However, it is now being phased out in many countries because it is an ozone-depleting substance. Several other chemicals such as ethyl formate, nitric oxide, and hydrogen sulfide are now used as alternative fumigants (*Kuchi & Sharavani, 2019*). The use of these chemicals, especially nitric oxide, requires a sophisticated system, which may not be accessible to small-scale farmers (*Kuchi & Sharavani, 2019*). Thus, more alternative fumigation treatments capable of suppressing anthracnose disease in fresh strawberry fruits during post-harvest storage, transportation, and market periods, are needed.

Over the years, much attention has been paid to the application of biological agents to control pests and pathogen-related diseases. This is known as biocontrol (*Liu, Yang & Simmons, 2016*). One biocontrol technique is the use of VOCs produced by microorganisms

to inhibit the growth of and infections by pathogens on the fruit surface during the post-harvest period. This is also called post-harvest biofumigation (*Kaddes et al., 2019*; *Chen et al., 2020*). Many studies have reported the successful use of VOCs produced by microorganisms, especially endophytic fungi, as biological agents to control fruit and vegetable diseases. Endophytic fungi are microorganisms asymptomatically colonizing and living in plant tissues, especially leaves, stems, and roots (*Wan et al., 2008*; *Morath, Hung & Ennett, 2012*). Some of the endophytic fungi, such as those of the *Daldinia* genus, are capable of producing bioactive compounds against a variety of phytopathogens. For instance, *Pandey & Banerjee (2014)* reported that the endophytic fungus *Daldinia bambusicola* Ch4/11 produced bioactive metabolites such as linalool, benzeneethanol, 2H-1-benzopyran-2-one, pivalic acid anhydride, and 2-ethylhexanol, which were able to suppress the growth of several plant pathogens, including *Colletotrichum lagenarium*. Similarly, the VOCs produced by the endophytic fungus *Daldinia cf. concentrica,* which included 3-methyl-1-butanol, (±)-2-methyl1-butanol, 4-heptanone, isoamyl acetate, and trans-2-octenal, were considered to possess antifungal activities against various plant pathogenic fungi (*Liarzi et al., 2016*). Other fungi that produced VOCs against plant pathogens include *Trichoderma* (*Valenzuela et al., 2015*), *Noduliosporium* (*Suwannarach et al., 2013*), and *Muscador albus* (*Mercier & Jiménez, 2004*). According to previous studies, endophytic fungi capable of producing volatile compounds that can be used for biofumigation include *Streptomyces globisporus* JK-1 (*Li et al., 2012*), *Trichoderma* (*Valenzuela et al., 2015*), *Noduliosporium* (*Suwannarach et al., 2013*), and *Muscador albus* (*Mercier & Jiménez, 2004*). The use of microorganism-derived VOCs for fumigation of fresh fruits is relatively inexpensive and may require less sophisticated systems, in comparison to the use of toxic chemicals. Thus, biofumigation can likely be performed by local farmers to extend the shelf-lives of their agricultural produce. It is also considered an environmentally- and user-friendly technique. Although the use of microorganism-derived VOCs as biofumigants shows a promising advantage in controlling post-harvest diseases, not many studies are available in the literature that present results on the use of microorganism-derived VOCs to prevent anthracnose diseases in fresh strawberry fruits (*Huang et al., 2012*; *Chen et al., 2020*; *Alijani et al., 2019*; *Zhang et al., 2015*). This warrants additional research studies that may lead to the successful use of microorganism-derived VOCs to control this disease.

*Barleria prionitis* (family Acanthaceae), known as an Ayurvedic herb, is endemic to tropical Asia and Africa (*Sahu, 1984*). It has been used as an herbal medicine to treat various health symptoms (*Talukdar, Rahman & Paul, 2015*). Previous studies reported that this plant species contains several secondary metabolites that have antimicrobial activities (*Amoo et al., 2011*; *Singh et al., 2005*). It was hypothesized that this plant is a host of endophytic fungi that are capable of producing VOCs that suppress post-harvest fungal infections in fresh fruits. Therefore, this study aimed to isolate endophytic fungi from *B. prionitis* leaves and investigate the antifungal properties of their VOCs against *C. acutatum* mycelial growth and its infection on harvested strawberry fruits. The results of this study may lead to an alternative method to control post-harvest anthracnose disease caused by *C. acutatum* on fresh strawberry fruits.

## MATERIALS & METHODS

### Plant material

*Barleria prionitis* plants, aged approximately 5 years, were collected from Chiang Rai province, Thailand in October 2019. Healthy *B. prionitis* leaves from the collected plants were sampled and sealed in plastic bags before being stored at 4 °C and used within 48 h of collection.

### Isolation of endophytic fungi

The leaves of *B. prionitis* were washed with tap water to eliminate soil, dust, and dirt attached to the surface and rinsed with distilled water 3 times. Surface of the leaves was sterilized by immersing them in 70% ethanol for 1 min, 5% sodium hypochlorite for 30 s, and 70% ethanol for 30 s, to remove epiphytic microorganisms. After this, the leaves were washed thoroughly using sterilized distilled water to remove remaining solvents. The surface of the leaves was dried with sterilized tissue paper to provide aseptic conditions. The dried leaves were cut with a sterilized scalpel blade into small pieces (i.e., approximately 0.5 cm × 0.5 cm segments). This surface sterilization procedure was performed following the study by *Monggoot et al. (2017)*.

Each leaf segment was placed on a Petri plate containing potato dextrose agar (PDA) medium mixed with 30 µg/mL of chloramphenicol to prevent bacterial growth. A few drops of sterilized distilled water from the final washing step were placed on PDA plates (n=5) to confirm that no contamination was observed (negative controls). All PDA plates were sealed with Parafilm. The PDA plates were incubated at 27 °C and examined daily for 7 days. Once the mycelia fully appeared, each hypha was further transferred onto a new, freshly prepared PDA plate. This procedure was repeated at least 3 times until a pure culture of each endophytic isolate was achieved. The endophytic isolates were coded as BP(x), where x was a numeric number assigned to each isolate.

### Isolation and molecular identification of phytopathogen *C. acutatum*

Phytopathogenic fungus *C. acutatum* was isolated from 10 infected strawberry fruits that were collected from a strawberry field located at Angkhang Royal Agricultural Station in Fang district, Chiang Mai province, Thailand in February 2020. Each infected fruit was washed with sterile distilled water before transferring to a Petri plate containing PDA mixed with 30 µg/mL of chloramphenicol. They were incubated at 27 °C for 7 days. All fungi obtained from the infected strawberry fruits were sub-cultured to obtain a pure culture. The isolated phytopathogen was pre-identified based on micro-morphological and macro-morphological characteristics, including morphology of sporulation. The aerial mycelium of *C. acutatum* isolate was scraped from the PDA surface and pulverized with a mortar and pestle to achieve a mycelium pulp. The genomic DNA was extracted following a protocol described by *Tanapichatsakul et al. (2020)*. The internal transcribed spacer (ITS) region was amplified using ITS5/ITS4 primers (*Tanapichatsakul et al., 2020*). Polymerase chain reactions (PCR) were performed following a protocol described by *Tanapichatsakul et al. (2020)*. The sequence similarity of isolated fungi from BLAST result was 100% to *C. acutatum*. In addition, the pathogenicity test of an isolated pathogen was also performed

using healthy strawberries to confirm pathogen identification. Ten strawberry fruits were wounded approximately 5 mm diameter and inoculated with a plug of *C. acutatum*. Fruits were kept in a moist plastic box and incubated at 27 °C for 7 days. Control fruits were inoculated with PDA agar plug. The test was performed in 3 replicates and repeated 3 times ($n = 90$). The symptoms and morphology of *C. acutatum* on strawberries were similar with previous studies (*Kumvinit & Akarapisan, 2016*; *Es-Soufi et al., 2018*).

## In vitro screening of antifungal activity of VOCs produced by isolated endophytic fungi on the radial growth of *C. acutatum*

The VOCs produced by each isolated endophytic fungus against *C. acutatum* were screened for antifungal activity using the dual-culture plate method described by *Li et al. (2015)*, with a slight modification on the incubation time. Two sterilized PDA bottom plates (9 cm diameter) were used. An agar plug (5 mm diameter, 7 days old) of *C. acutatum* was placed at the center of one plate, whereas an agar plug (5 mm diameter, 7 days old) of the individual endophytic isolate was placed at the center of another plate. The two plates were sealed together and wrapped with Parafilm before incubation at 27 °C for 7 days. Plates without the endophytic isolate but with *C. acutatum* were used as controls. The experiment was performed using 5 replicates for each endophytic isolate and repeated 3 times ($n = 15$). The mycelial diameter of *C. acutatum* in each treatment was measured daily using a vernier digital caliper (MITUTOYO-ABS Digimatic Caliper CD-AX, Japan) in a millimeter unit. The obtained results were expressed as inhibition percentage of mycelial growth according to the following formula:

$$\% \text{ Inhibition} = \left( \frac{\text{mycelial diameter of untreated fungus} - \text{mycelial diameter of treated fungus} \times 100}{\text{mycelial diameter of untreated fungus}} \right).$$

## Viability test of the treated *C. acutatum*

From the in vitro screening above, the treatment that exhibited the highest antifungal activity was selected for a viability test of the treated *C. acutatum*. An agar plug (5 mm diameter) of *C. acutatum* treated with the chosen endophytic isolate was placed on a fresh PDA plate and further cultured at 27 °C for 7 days to evaluate its viability. The mycelial growth of *C. acutatum* was evaluated. The experiment was performed using 5 replicates and conducted 3 times ($n = 15$).

## In vivo antifungal activity of the VOCs produced by the endophytic isolate BP11 against *C. acutatum* infections on strawberry fruits

The in vivo experiment was performed using healthy strawberry fruits collected in March 2020. All fruits were selected according to their maturity, size, color, and the absence of physical injuries or infections. The fruits were harvested early in the morning and then transported to the laboratory within 3 h. The fruit surfaces were sanitized using 1% sodium hypochlorite for 5 min, then rinsed with distilled water 3 times before being placed on a sterilized tissue paper for aseptic drying.

Inoculation of strawberry fruits with *C. acutatum* was performed according to the procedure described by *Li et al. (2015)*, with a slight modification. One location on each cleaned strawberry fruit was wounded using a sterile inoculating needle. The wound depth was approximately 5 mm. A plug of *C. acutatum* (5 mm diameter, 7 days old) was placed into the wound. After inoculation, the strawberry samples were placed inside a plastic box (15.5 cm ×13.5 cm ×7.0 cm). The bottom of the plastic box was covered with sterilized medical gauze. Double layers of autoclaved tissue paper soaked in 100 mL of sterile distilled water were placed underneath the gauze to provide moisture. Then, a 7-day old endophytic isolate BP11 culture plate, without its cover, was placed in the same box, but without direct contact with the strawberry samples. Each box was closed with a fitted plastic lid and air-sealed using Parafilm. All boxes were stored at 27 °C for 7 days. To evaluate the effect of VOCs on the growth of *C. acutatum*, 4 different treatments were prepared, including pathogen-inoculated fruits in the plastic box (positive control), pathogen-uninoculated fruits in the endophytic isolate BP11-containing plastic box (negative control), pathogen-uninoculated fruits in the plastic box (control), and pathogen-inoculated fruits in the endophytic isolate BP11-containing plastic box (BP11 treatment). There were 15 strawberry samples in each box. Each treatment was performed in 3 replicates and repeated 3 times ($n = 135$). The mycelial diameter of *C. acutatum* on strawberries from all treatments was measured using a vernier digital caliper in a millimeter unit. In this experiment, the inhibition percentage was calculated as explained above. Disease incidence and disease severity percentages were calculated using the following formulas:

$$\% \text{ Disease incidence} = \left( \frac{\text{number of disease fruits}}{\text{total number of fruits assessed}} \right) \times 100$$

$$\% \text{ Disease severity} = \left( \frac{\text{sum of all disease rating}}{\text{total number of rating } \times \text{ maximum disease scale}} \right) \times 100.$$

For the disease severity percentage, the rotten area on the surface of each strawberry fruit was visually evaluated using a scale from 0 to 5. A description for each scale is as follows: 0 = no disease symptom, 1 = a small grey spot covering <1% fruit area, 2 = a grey sunken spot covering 1–10% fruit area, 3 = a grey spot covering 11–25% fruit area, 4 = circular grey sunken spots covering 26–50% fruit area, and 5 = circular to irregular spots covering >51% fruit area.

## Quality of strawberry fruits after treating with VOCs produced by the chosen endophytic isolate (BP11)

The quality of strawberry fruits was evaluated after exposure to VOCs produced by the endophytic isolate BP11 for 7 days. In this experiment, fruit firmness, total soluble solids, and pH were measured following the method of *Cai et al. (2015)*. Fruit firmness was measured by a TA-XT2i texture analyzer (Stable Micro Systems Ltd., UK) with a P50 cylinder plunger probe at the equator of the equidistant region. Fruit firmness was recorded from the maximum force. The texture analyzer was adjusted following this condition: 5.0 mm/s pre-test speed, 1.0 mm/s test speed, and 5.0 mm/s post-test speed, all with a penetration distance of 5 mm. Strawberry juice was extracted and filtered through

two layers of cheesecloth. Total soluble solids of the strawberry fruit juice were measured using a hand-held refractometer (WYT-4) (Top Instrument Co., Ltd., China). The pH value was also determined by a pH meter (DELTA 320) (Top Instrument Co., Ltd., China). In this experiment, all treated strawberry samples from above were used.

## Identification and analysis of VOCs produced by the endophytic isolate BP11

The VOCs produced by the endophytic isolate BP11 were extracted using solid phase micro-extraction (SPME). Prior to extraction, the endophytic isolate BP11 was cultured in a glass container (9 cm diameter) containing PDA medium at 27 °C for 7 days and covered with an aluminum cap with a PTFE-coated silicone septum. The SPME sampling apparatus with a SPME holder and a 1.0 cm fused silica fiber was purchased from Supelco (PA, USA). A 50/30 μm divinylbenzene-carboxen-polydimethylsiloxane (DVB-CAR-PDMS) SPME fiber was selected to extract the VOCs. Before extraction, the SPME fiber was preconditioned at 230 °C inside the injection port of a gas chromatograph (HP 6890) (Agilent Technologies, CA, USA) for 30 min. The preconditioned SPME fiber was then inserted into the glass container containing the endophytic isolate BP11 to trap the VOCs for 30 min prior to immersion in the injection port of the HP model 6890 gas chromatograph coupled to an HP model 5973 mass-selective detection instrument (Agilent Technologies) for 10 min in splitless mode. An HP-5ms (30 m ×0.25 mm i.d., 0.25 μm film thickness) column (Agilent Technologies) was used for compound separation. The oven temperature was initially set to 40 °C and then increased at a rate of 2 °C/min to the final temperature of 200 °C. The injector temperature was set to 250 °C. Helium was used as a carrier gas and was set to a flow rate of 1 mL/min. The mass spectrometer was operated using electron impact ionization (70 eV electron ionization voltages). A scan mode (cover the range of m/z 30-300) was used. The ion chamber and the mass transfer temperatures were set to 250 °C and 230 °C, respectively.

The $C_9$-$C_{17}$ n-alkanes were used to calculate the retention time index. Mass spectra stored in NIST05 Library, Wiley7N Mass Spectral Library, and *Adams* Library *(2017)* were used for compound identification. A gas chromatograph (HP 6890, Agilent Technologies) equipped with a flame ionization detector was used for quantitative analysis of the VOCs. The identified volatile constituents were reported as the percentage of relative peak areas calculated using the normalization method without correction factors.

## In vitro antifungal activity of synthetic VOCs on *C. acutatum*

Synthetic volatile standards of 3,5-dimethyl-4-heptanone (95%), ethyl sorbate (95%), benzaldehyde dimethyl acetal (95%), trans-sabinene hydrate (95%), methyl geranate (90%), and elemicin (95%) were used in this experiment. Ethyl sorbate, benzaldehyde dimethyl acetal, methyl geranate, and elemicin were purchased from Santa Cruz Biotechnology (TX, USA) while trans-sabinene hydrate was purchased from Sigma-Aldrich (China). These volatile standards were in a liquid form. The antifungal activity of each synthetic volatile compound was investigated according to the method of *Gotor-Vila et al. (2017)*, with a slight modification. In this experiment, 2 sterilized PDA bottom plates

(9 cm diameter) were used. An agar plug (5 mm diameter, 7 days old) of *C. acutatum* was placed at the center of one plate, whereas a paper filter (90 mm diameter) spiked with an aliquot of the volatile standard was placed at the center of another plate. Both plates were sealed together and wrapped with Parafilm before incubation at 27 °C for 7 days. Each synthetic VOC was tested separately using 3 different aliquots (12.5, 25.0, and 50.0 μL) of its liquid solution. Because the Petri plates have a remaining headspace volume of 44 mL, the corresponding concentrations of the synthetic compound were 0.28, 0.56, and 1.12 μL/mL headspace, respectively. Petri plates without synthetic VOCs but with *C. acutatum* were used as controls. For each synthetic volatile compound, 5 replicates for each spiking volume were used and repeated 3 times ($n = 45$ for all 3 levels). The mycelial diameter of *C. acutatum* in each treatment was measured using a vernier digital caliper in a millimeter unit. The obtained results were expressed as inhibition percentage of mycelial growth as explained above.

### In vivo antifungal activity of elemicin against *C. acutatum* infections on strawberry fruits

Strawberry fruits were prepared and wounded as explained above. They were placed at the bottom of the plastic box (15.5 cm ×13.5 cm ×7.0 cm) prepared in a similar manner as explained above. Then, a Petri plate containing 3 paper filters (90 mm diameter), each moistened with 50 μL of the elemicin standard, was placed in the same box, but without direct contact with the strawberry samples. Each box was closed with a fitted plastic lid and air-sealed using Parafilm. All boxes were stored at 27 °C for 7 days. Treatment that included only the pathogen-inoculated fruits in the plastic box was used as a positive control. In each box, there were 15 strawberry samples. Each treatment was performed in 3 replicates and repeated 3 times ($n = 135$). The mycelial diameter of *C. acutatum* on strawberry samples from all boxes was measured using a vernier digital caliper in a millimeter unit. The inhibition, disease incidence, and disease severity percentages were calculated as described above.

### Quality of strawberry fruits after treating with elemicin

The quality of strawberry fruits was evaluated after treating with elemicin. This evaluation was done similarly to the procedure outlined above.

### Molecular identification of the endophytic isolate BP11

Genomic DNA of endophytic isolate BP11 was extracted using a Genomic DNA Extraction Mini-Kit following the method of *Tanapichatsakul et al. (2020)*. DNA loci including ITS, LSU, and RPB2 regions were amplified by PCR following the protocol of *Samarakoon et al. (2019)*. The obtained sequences in this study were subjected to BLAST search in GenBank (https://blast.ncbi.nlm.nih.gov/Blast.cgi). BLAST search indicated that the isolate BP11 belongs to Hypoxylaceae taxa. The sequences of representative Hypoxylaceae taxa used in the phylogenetic analyses were chosen from GenBank based on the BLASTn searches and recently published data (*Samarakoon et al., 2019*). Sequence datasets from all fungi obtained from ITS, LSU, and RPB2 rDNA primers were used to construct the phylogenetic tree. The accession numbers of all strains are shown in Table S1. The combined sequence

data were first aligned using MAFFT version 7 (http://mafft.cbrc.jp/alignment/server/). The alignment was further improved using BioEdit v. 7.0.5.3. The last alignment of the combined sequence datasets was investigated and the phylogenetic tree inferred based on maximum likelihood (ML) and Bayesian inference analyses (BI). ML and BI analyses were achieved using the RAxML-HPC2 on XSEDE (v. 8.2.12) via the CIPRES Science Gateway platform and MrBayes on XSEDE, MrBayes 3.2.6 via the CIPRES Science Gateway platform, respectively. Bayesian posterior probabilities (PP) were investigated using Markov Chain Monte Carlo sampling (BMCMC). The phylogenetic tree was constructed in FigTree v.1.4.3. The endophytic isolate BP11 was deposited in the Mae Fah Luang University Culture Collection (MFLUCC), Chiang Rai, Thailand under the accession number MFLUCC 19-0493.

### Analysis of data

Data from each experiment were presented as the mean ± standard deviation. When applicable, data were subjected to Analysis of Variance (ANOVA), followed by *post-hoc* multiple pairwise comparisons using Duncan's multiple range tests ($\alpha = 0.05$) or Dunnett comparisons with a reference treatment ($\alpha = 0.05$). Linear regression analysis was conducted to evaluate the relationship between the observed inhibition percentage and the concentrations of synthetic VOCs tested. The statistical analyses were conducted using the SPSS 20.0 software (IBM Corp; 2011, NY, USA).

## RESULTS

### In vitro antifungal activity of VOCs produced by endophytic isolates and viability of the treated *C. acutatum*

A total of 34 endophytic isolates were obtained from the *B. prionitis* leaves. After 7-day incubation, only 26 of them (Fig. S1) were able to produce VOCs that inhibit the mycelial growth of *C. acutatum,* but to varying degrees (Fig. 1). For these endophytic isolates, the average inhibition percentages ranged between 3.3 and 80.8%. The endophytic isolate BP11 was found to produce the VOCs with the highest inhibition percentage (80.8, $p < 0.05$). Moreover, the in vitro antifungal activity of the VOCs produced by the endophytic isolate BP11 on *C. acutatum* across 7 days of the incubation period was depicted in Fig. S2. The highest antifungal activity was observed from day 4 to day 7. During which time, the average inhibition rate was about 80%. From the viability test, *C. acutatum* treated with the VOCs produced from the endophytic isolate BP11 could not reproduce its mycelia on the PDA plates over a 5-day incubation period (data not shown).

### In vivo antifungal activity of VOCs produced by the endophytic isolate BP11 against *C. acutatum* infections on strawberry fruits

Strawberry samples after undergoing various treatments are shown in Fig. 2. Physical damage on the strawberry fruits was neither detected in the pathogen-uninoculated fruits containing the endophytic isolate BP11 (negative control) nor the pathogen-uninoculated fruits (control). In comparison to pathogen-inoculated fruits (positive control) (Fig. 2C), less physical damage was observed on fruits inoculated with *C. acutatum* and treated with the VOCs produced by the endophytic isolate BP11 (Fig. 2D).

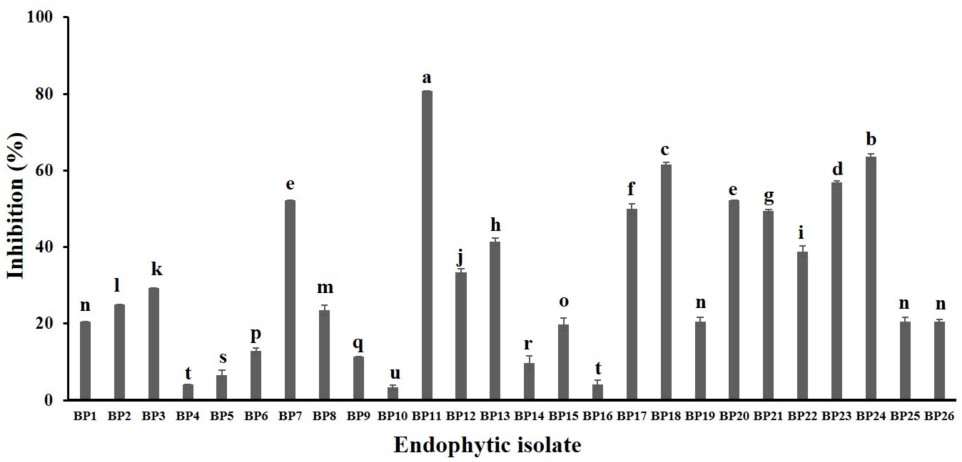

**Figure 1** In vitro antifungal screening of the endophytic isolates against the mycelial growth of *C. acutatum* (measured on day 7 of the incubation period). The data for each bar are mean ± standard deviation ($n = 15$ per isolate). Different letters above the bars indicate significant differences ($p < 0.05$) among the observed inhibition percentages (ANOVA, followed by Duncan's multiple range test).

## VOCs produced by the endophytic isolate BP11

Gas chromatographic-mass spectrometric (GC-MS) analysis of the VOCs produced by the endophytic isolate BP11 led to the identification of 60 compounds. These VOCs are listed in Table 1. The volatile profile of the endophytic isolate BP11 was dominated by the presence of hydrocarbons (35.0%), followed by monoterpene hydrocarbons (39.4%), and sesquiterpene hydrocarbons (37.3%). The major volatile compounds produced by the endophytic isolate BP11 were elemicin (23.8%), benzaldehyde dimethyl acetal (8.5%), ethyl sorbate (6.8%), methyl geranate (6.5%), trans-sabinene hydrate (5.4%), and 3,5-dimethyl-4-heptanone (5.1%). In addition, the VOCs produced from the endophytic isolate BP11 possessed sweet and fruity odor.

## In vitro antifungal activity of synthetic VOCs against *C. acutatum*

The in vitro antifungal activities of the 6 most abundant VOCs against *C. acutatum* are summarized in Fig. 3. The results show that these chosen synthetic VOCs had varying degrees of antifungal activity. For every volatile compound tested, the highest inhibition percentage was detected on day 1 and continued to decrease in the later days ($p < 0.05$). Elemicin demonstrated the highest inhibition percentages across the observation period, covering a range of 15.3% to 76.6% ($p < 0.05$) (Fig. 3). Ethyl sorbate showed poor antifungal activity against *C. acutatum*.

Positive associations were found between the amounts of synthetic VOCs used (per mL headspace) and the inhibition percentages. In the case of elemicin, the resulting linear regression slope on day 1 of the incubation period was 22.1 (Fig. S3). Across the observation period, the slopes of elemicin were in a range of 22.1 to -24.7. Moreover, the linear regression slopes of other compounds (on day 1) was also positive with a value of 22.5, 18.0, 13.4,

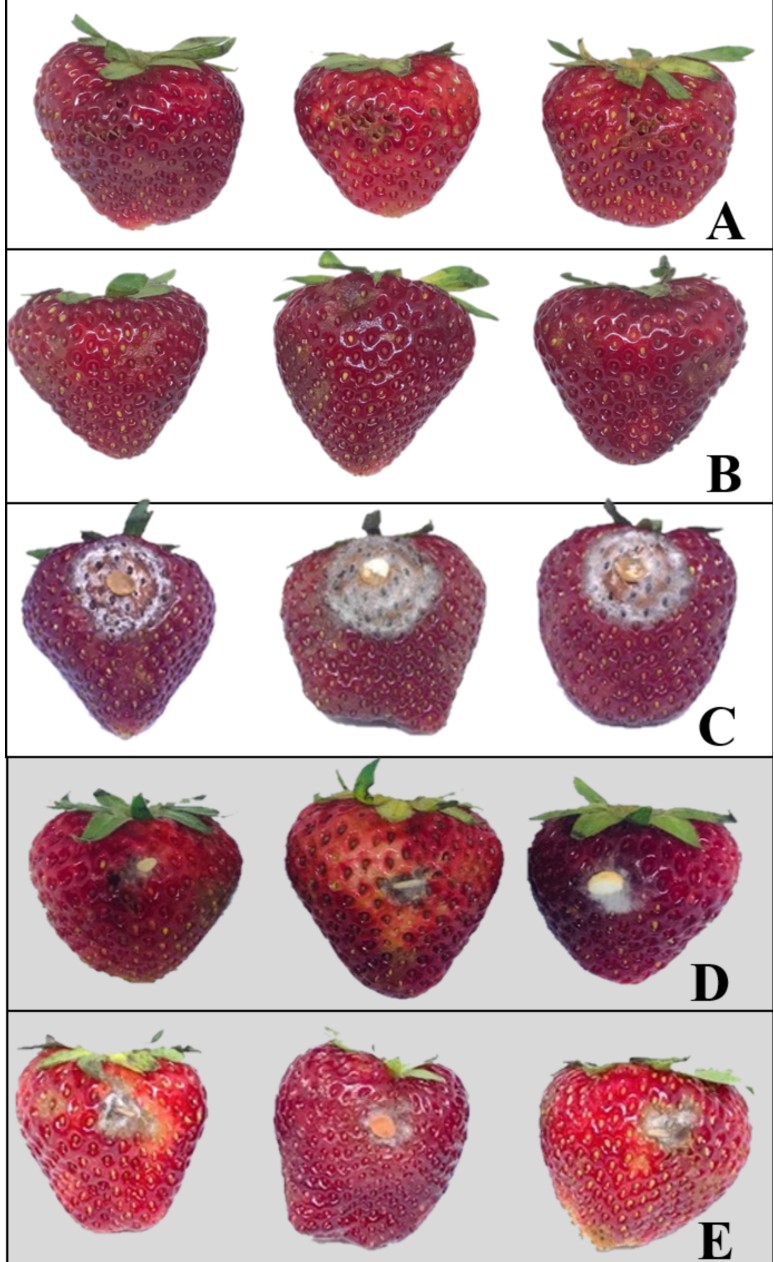

**Figure 2  Examples of strawberries obtained from various in vivo experiments; negative control (A), control (B), positive control (C), endophytic isolate BP11 (D), and elemicin (E).** Each treatment was performed in three replicates of ($n = 15$) and repeated three times (total $n = 135$).

**Table 1   Identified volatile compounds from the endophytic fungus *D. eschscholtzii* MFLUCC 19-0493 analyzed by SPME-GC-MS.**

| Compound | Retention index | % area | Compound | Retention index | % area |
|---|---|---|---|---|---|
| 3,5-dimethyl-4-heptanone | 973 | 5.1 | 2-adamantanone | 1310 | 1.0 |
| octen-3-ol | 974 | 0.1 | methyl geranate | 1322 | 6.5 |
| 2-octanone | 988 | 0.3 | anisyl formate | 1330 | 1.0 |
| 2,4,5-trimethyl-thiazole | 994 | 1.5 | 4-hydroxy-benzenemethanol | 1335 | 0.2 |
| 2-ethyl-3-methyl-pyrazine | 1002 | 1.1 | silphinene | 1345 | 0.4 |
| iso-sylvestrene | 1007 | 3.0 | 9-decenoic acid | 1359 | 0.5 |
| 2-acetyl-thiazole | 1014 | 3.4 | para-methyl anisate | 1371 | 1.4 |
| sylvestrene | 1025 | 0.7 | thujic acid | 1382 | 1.0 |
| cis-arbusculone | 1046 | 0.2 | alpha-duprezianene | 1387 | 1.1 |
| meta-tolualdehyde | 1069 | 0.2 | alpha-thujaplicin | 1410 | 0.1 |
| 2E,4E-hexadienol acetate | 1081 | 3.0 | ethyl-anthranilate | 1414 | 0.3 |
| diethyl acetal-2Z-hexenal | 1085 | 3.1 | beta-duprezianene | 1422 | 0.2 |
| 2-acetyl thiophene | 1086 | 0.7 | dictamnol | 1428 | 0.1 |
| ethyl sorbate | 1092 | 6.8 | pinonic acid | 1440 | 0.3 |
| methylthiopyrazine | 1093 | 0.6 | prezizaene | 1444 | 0.3 |
| trans-sabinene hydrate | 1098 | 5.4 | alpha-acoradiene | 1464 | 0.5 |
| benzaldehyde dimethyl acetal | 1109 | 8.5 | gamma-muurolene | 1478 | 0.3 |
| 2-ethyl hexanoic acid | 1119 | 0.5 | isodaucene | 1500 | 0.2 |
| trans-pinene hydrate | 1119 | 2.0 | alpha-thujaplicinol | 1509 | 0.7 |
| alpha-campholenal | 1122 | 0.3 | trans-calamenene | 1521 | 0.2 |
| stemone | 1124 | 0.2 | zonarene | 1528 | 0.4 |
| Z-myroxide | 1131 | 0.5 | cis-calamenene | 1528 | 0.1 |
| iso-3-thujanol | 1134 | 1.9 | elemicin | 1555 | 23.8 |
| 2-acetyl-3-ethyl-pyrazine | 1156 | 0.3 | E-nerolidol | 1561 | 0.3 |
| fragranol | 1214 | 0.6 | Z-asarone | 1616 | 0.5 |
| nor-davanone | 1228 | 0.4 | epi-cedrol | 1618 | 0.1 |
| benzene acetic acid ethyl ester | 1243 | 0.9 | 5-cedranone | 1628 | 0.1 |
| Z-anethole | 1249 | 1.6 | cedr-8(15)-en-10-ol | 1650 | 0.1 |
| methyl nerolate | 1280 | 1.8 | 5-iso-cedranol | 1672 | 0.2 |
| 2E,6Z-nonadienol acetate | 1303 | 0.6 | 5-neo-cedranol | 1684 | 0.6 |

**Notes.**
Retention index was determined using the homologous series of n-alkanes. The % peak area was calculated using the normalization method without correction factors.

22.3, and 8.9 for 3,5-dimethyl-4-heptanone, ethyl sorbate, benzaldehyde dimethyl acetal, trans-sabinene hydrate, and methyl geranate, respectively (data not shown).

## In vivo antifungal activity of elemicin against *C. acutatum* infections on strawberry fruits

Strawberry fruits after treatment with elemicin are shown in Fig. 2E. Less physical damage was observed on the fruits inoculated with *C. acutatum* and treated with elemicin than the positive control samples. The inhibition, disease incidence, and disease severity percentages of this treatment were 70.9%, 21.5%, and 15.9%, respectively (Table 2). These results were similar to those obtained from the endophytic isolate BP11 treatment.

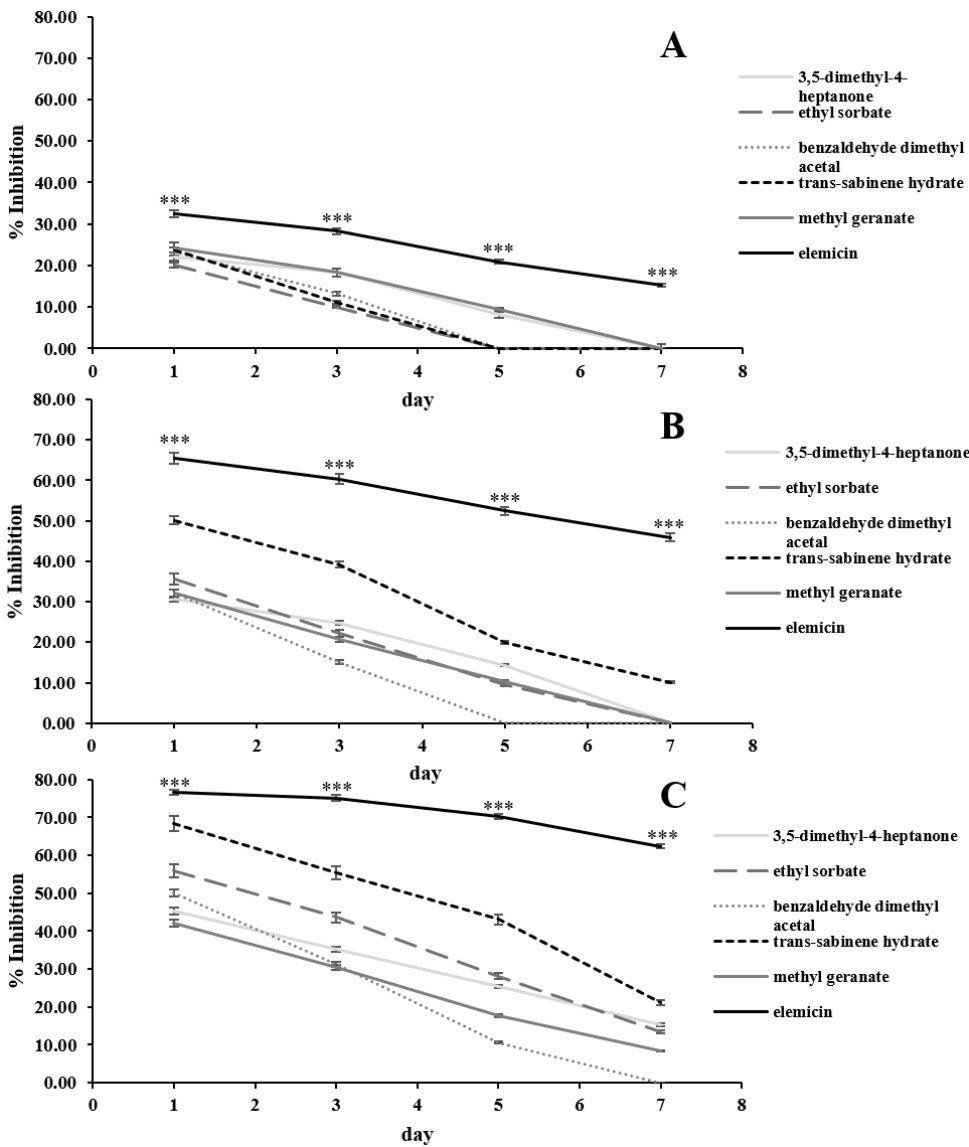

**Figure 3** In vitro antifungal activity of synthetic volatile compounds at the starting concentration of 0.28 (A), 0.56 (B), and 1.12 µL/mL headspace (C) on *C. acutatum* during a 7-day incubation period. The data for each point are mean ± standard deviation ($n = 15$ per day). The asterisks (***) above the points indicates a significant difference ($p < 0.001$) in the inhibition percentage between elemicin and other compounds observed in each day (ANOVA, followed by Dunnett's multiple range test).

## Strawberry quality after treating with VOCs produced by the endophytic isolate BP11 or elemicin

The quality of strawberry fruits obtained from various treatments, including the synthetic compound elemicin, in terms of firmness, total soluble solids, and pH are demonstrated in Table 3. After 7 days of treatment, the average firmness of pathogen-inoculated fruits (positive control) was 1.2 N. It was significantly lower than other treatments. The average firmness of pathogen-inoculated fruits (positive control), pathogen-uninoculated fruits

**Table 2  Inhibition, severity, and incidence percentages of *C. acutatum* in strawberries treated with *D. eschscholtzii* MFLUCC 19-0493 and elemicin observed from the in vivo experiments.**

| Treatment | % inhibition | % severity | % incidence |
|---|---|---|---|
| *D. eschscholtzii* MFLUCC 19-0493 | 72.4 ± 0.9 | 19.9 ± 0.8 | 15.6 ± 0.7 |
| elemicin | 70.9 ± 1.2 | 21.5 ± 0.6 | 15.9 ± 0.9 |

Notes.
The data are mean ± standard deviation (total $n = 135$).

**Table 3  Quality of fresh strawberries measured from the *in vivo* assays.**

| Treatment | Fruit quality | | |
|---|---|---|---|
| | Fruit firmness ($N$) | Total soluble solids (%) | pH |
| **BP11** | **3.9 ± 0.0[a]** | **7.8 ± 0.1[a]** | **3.3 ± 0.0[a]** |
| **Elemicin** | **3.9 ± 0.1[a]** | **7.7 ± 0.1[a]** | **3.3 ± 0.0[a]** |
| Control | 3.9 ± 0.3[a] | 7.9 ± 0.0[a] | 3.3 ± 0.0[a] |
| Negative control | 3.8 ± 0.3[a] | 7.9 ± 0.1[a] | 3.4 ± 0.0[a] |
| Positive control | 1.2 ± 0.8[b] | 7.7 ± 0.1[a] | 3.4 ± 0.0[a] |

Notes.
[a,b] The data are mean ± standard deviation ($n = 135$, per treatment). Different letters indicate significant differences ($p < 0.05$) among values within each category (ANOVA, followed by Duncan'ss multiple range test).

containing the endophytic isolate BP11 (negative control), pathogen-uninoculated fruits (control), pathogen-inoculated fruits containing the endophytic isolate BP11 (BP11 treatment), and pathogen-inoculated fruits containing elemicin (elemicin treatment) ranged from 3.8 to 3.9 N. These values were not significantly different. No significant difference was observed for the total soluble solids content (7.7–7.9) and the pH (3.3–3.4) of the strawberry fruits from all treatments.

## Phylogenetic tree and molecular identification

The phylogenetic tree of the endophytic isolate BP11 is shown in Fig. 4. Based on the results, the endophytic isolate BP11 was identified as *Daldinia eschscholtzii* MFLUCC 19-0493. Its genetic sequence was submitted to the GenBank database under the accession numbers MN704648, MW485486, and MW495050 from ITS, LSU, and RPB2 regions, respectively.

## DISCUSSION

The VOCs produced by the *D. eschscholtzii* MFLUCC 19-0493 possessed sweet and fruity odor, which is a feature of *Daldinia* fungi reported by *Stadler et al. (2014)*. These VOCs were able to significantly inhibit the mycelial growth of *C. acutatum* without affecting the firmness, total soluble solids, and pH of the strawberry fruits, as shown in Table 3. Thus, it is likely that fumigation of harvested strawberry fruits with the VOCs produced from the *D. eschscholtzii* MFLUCC 19-0493 will result in a prolonged storage time of harvested strawberry fruits by reducing the incidence and severity of the anthracnose disease caused by *C. acutatum*. The VOCs produced by the *D. eschscholtzii* MFLUCC 19-0493 is complex and comprised with at least 60 compounds. In comparison to other major compounds identified, elemicin possessed the strongest antifungal activity. The inhibition percentage

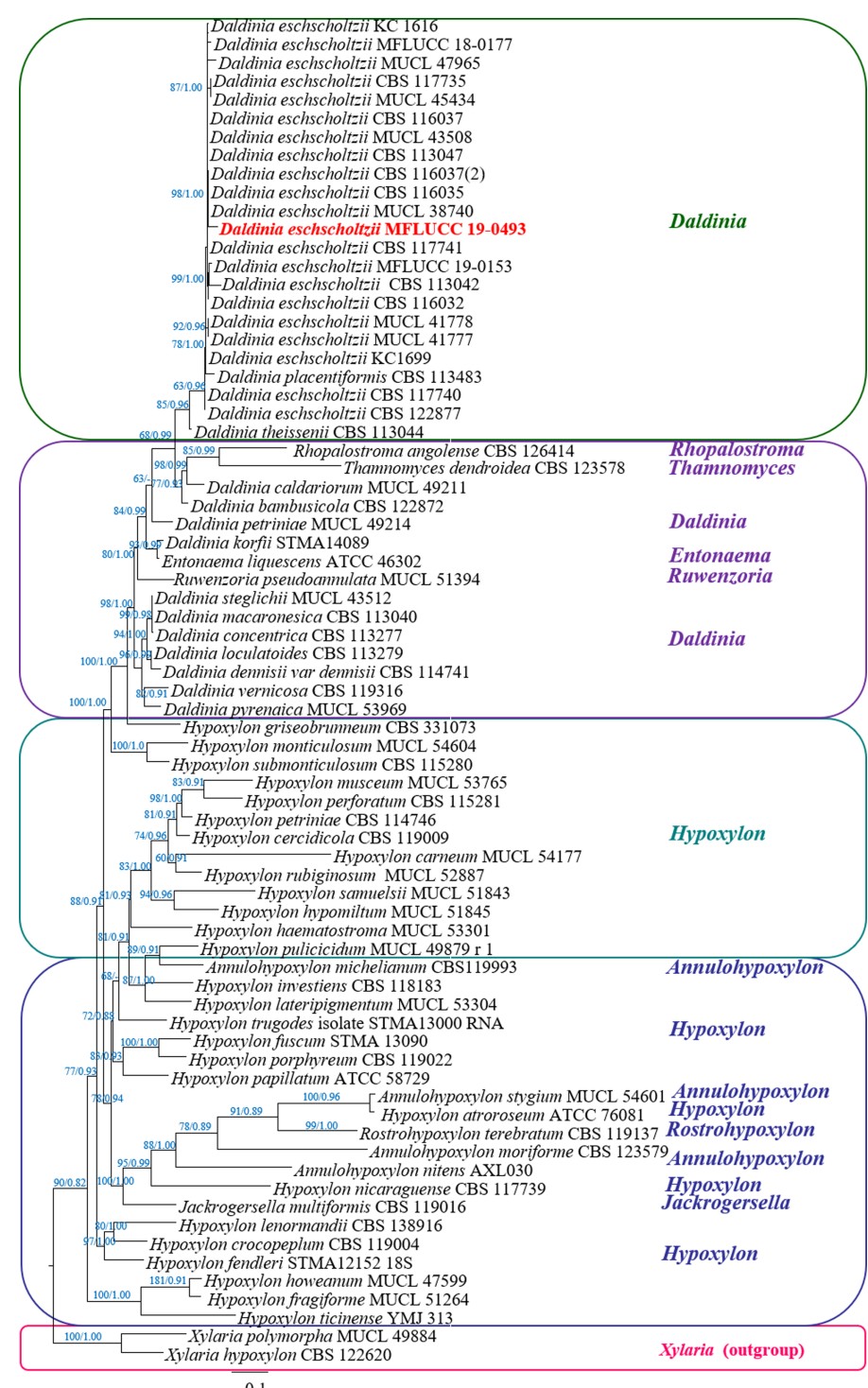

**Figure 4** Phylogram generated from RAxML analysis based on a combined LSU-ITS-RPB2 rDNA sequence data of *Daldinia*, showing the phylogenetic position of *D. eschscholtzii* MFLUCC 19-0493. Bootstrap support values for ML equal to or greater than 60%, and Bayesian posterior probabilities (PP) equal to or greater than 0.90 are defined as ML/PP above the nodes.

of elemicin increased as the concentration increased (Fig. 3). This suggests that the ability for elemicin to suppress the mycelial growth of *C. acutatum* is dose-dependent. The same trend was observed for other synthetic VOCs tested.

The results of this study, regarding the antifungal activity of elemicin, agreed with previous studies, which also demonstrated that elemicin was a key antifungal volatile compound. In these studies, elemicin was found to be able to inhibit the growth of *C. nymphaeae*, *C. gloeosporoides,* and *C. musae* (*Rossi et al., 2007*; *Karimi et al., 2016*). In addition, other VOCs such as trans-sabinene hydrate (*Btisam et al., 2016*), ethyl sorbate (*Nemes et al., 2020*), and benzaldehyde dimethyl acetal (*Goutam et al., 2016*) were all reported to possess some degree of antimicrobial activity. The synergistic or additive actions of these compounds may increase the antifungal activity of the VOCs produced by the *D. eschscholtzii* MFLUCC 19-0493. The precise inhibitory mechanism of VOCc against *C. acutatum* needs to be elucidated through further study. To the best of our knowledge, this is the first time that the antifungal activity of elemicin was demonstrated against *C. acutatum*. According to the study of *Tisserand & Young (2014)*, elemicin posed a negligible risk of carcinogenicity in humans at low doses. Thus, it has the potential to be used as a fumigant to control for *C. acutatum* infections on fresh strawberries. However, the production and ecological significance of elemicin from the *D. eschscholtzii* MFLUCC 19-0493 is not well understood and needs further investigation.

As observed from the in vitro screening experiment, the inhibitory patterns were different between the VOCs produced by the *D. eschscholtzii* MFLUCC 19-0493 and the selected synthetic VOCs. The observed inhibitory pattern of the synthetic VOCs was similar to the study of *Gao et al. (2018)* that the highest antifungal activity was present on the first day and decreased significantly on later days. The reduction in the inhibition percentage of the synthetic VOCs may be a result of a loss of chemical equilibrium or degradation from oxidation reaction. It is more likely that several volatile constituents contribute to the unique antifungal activity of the VOCs produced from *D. eschscholtzii* MFLUCC 19-0493. A few studies were able to propose the mechanisms by which certain VOCs inhibit the growth of pathogens. For instance, a combination of terpenoid compounds was found to inhibit the mycelial growth of selected pathogens by reducing the microbial cellular metabolic rates (*Serrano et al., 2005*; *Regnier et al., 2010*). Reduction of the metabolic rates may affect pathogenic microbial cell permeability and cell growth via reduction of the cell structure and its core function, causing the loss of macromolecules from the cell interior (*Tian et al., 2012*). In addition, the antifungal mechanisms of VOCs may depend on their ability to induce defense enzymes in the host rather than exerting the direct effect on the pathogens, as suggested by *Zhang et al. (2015)*.

Overall, the results of this study are in agreement with previous studies that reported the antifungal activity of the VOCs produced from *Daldinia* fungi. For instance, *Liarzi et al. (2016)* found that the VOCs produced by *Daldinia cf. concentrica* were able to control various plant pathogenic fungi. The major VOCs produced by *D. cf. concentrica* included trans-2-octenal, 1-methyl-1,3-cyclohexadiene, 2,2,5-trimethylcyclopentanone, 1-methyl-1,4-cyclohexadiene, and octanal. However, the antifungal VOCs were 3-methyl-1-butanol, (±)-2-methyl-1-butanol, 4-heptanone, isoamyl acetate, and trans-2-octenal. The VOC

profile obtained from *D. cf. concentrica* is substantially different from the one produced by *D. eschscholtzii* MFLUCC 19-0493. According to a previous study, some of the *Daldinia* fungi were reported to produce VOCs such as α-guaiene, guaia-1(10),11-diene, (-)- α-panasinsene, and thujopsene, which were identified as antimicrobial agents (*Pažoutová et al., 2013*). Also, the VOCs produced by *Hyproxylon anthochroum*, which is a member of the *Daldinia* genus, had strong antifungal activity against the growth of *Fusarium oxysporum* (*Macías-Rubalcava et al., 2018*). Moreover, 5-hydroxy-4,6-dimethyloctan-3-one isolated from *Daldinia clavata* as well as 5-hydroxy-2-methyl-4-chromanone isolated from *Daldinia cf. childiae* were evaluated to possess a high degree of antimicrobial activity (*Wang et al., 2018*; *Lauterbach et al., 2019*).

Due to the chosen extraction technique, some volatile compounds, such as small polyketides, which are mainly produced by the *Daldinia* fungi, could not be detected. These compounds are highly polar and were also reported to have antifungal activity (*Liarzi et al., 2016*). Our study is among just a few studies that have evaluated the use of endophytic fungal-derived VOCs as potential biofumigants to control for *C. acutatum* infections on strawberry fruits. A very recent study, conducted by *Alijani et al. (2019)*, reported that the VOCs produced from the bacterial isolate *Staphylococcus sciuri* MarR44 possessed both in vitro and in vivo antifungal activities against *C. nymphaeae*, another pathogen that is responsible for anthracnose disease in fresh strawberry fruits. Our study is different from the study by *Alijani et al. (2019)* in terms of the selected pathogen and the target endophytic organism.

Our study suggests that VOCs from *D. eschscholtzii* MFLUCC 19-0493 could be used as fumigation agents to control the *C. acutatum* associated anthracnose disease by inhibiting the colonization of *C. acutatum* on fresh strawberry fruits during the post-harvest period. These VOCs, including the major compound, elemicin, are not phytotoxic to strawberry fruits. It is likely that small-scale farmers can use these VOCs to effectively control for this type of anthracnose disease. However, in order to utilize *D. eschscholtzii* MFLUCC 19-0493, there is a need to establish a sustainable condition to support the continued growth of the fungus. Further studies are needed to optimize the culture and to effectively maintain life-supporting conditions for *D. eschscholtzii* MFLUCC 19-0493 in order to enhance the antifungal efficiency of its producing VOCs. Determination of the optimum concentration of the VOCs produced by the *D. eschscholtzii* MFLUCC 19-0493 is also necessary for their effective use in controlling infections of *C. acutatum* on strawberries during the post-harvest period. Lastly, more studies are needed to confirm the effectiveness of elemicin against *C. acutatum*.

## CONCLUSIONS

The present study showed that the VOCs produced by the *D. eschscholtzii* MFLUCC 19-0493 exhibited significant antifungal activity against the phytopathogen *C. acutatum*, which is usually responsible for anthracnose disease on fresh strawberries, without causing changes in firmness, total soluble solids, and pH of the fruits. Based on the GC-MS analysis, there were at least 60 VOCs produced by the *D. eschscholtzii* MFLUCC 19-0493. The major VOCs

were elemicin, benzaldehyde dimethyl acetal, trans-pinene hydrate, and 2-adamantanone. From the screening assay, elemicin is likely the key VOCs that provides the majority of antifungal effect. Overall, the volatile compounds produced by the *D. eschscholtzii* MFLUCC 19-0493 may be used as biofumigation agents to control post-harvest infections of *C. acutatum*. The use of these VOCs for fumigation of fresh strawberries is relatively inexpensive. They may be used by local farmers and could replace synthetic chemicals or toxic gases that are currently used as fumigants to control the post-harvest anthracnose disease caused by *C. acutatum*.

## ACKNOWLEDGEMENTS

The authors would like to thank the Center of Excellence in Fungal Research, Mae Fah Luang University for their support. We appreciate Dr. Sakon Monggoot for his support on the molecular analysis and fungal identification.

### Funding

This work was financially supported by the Mae Fah Luang University grant and the Royal Golden Jubilee Ph.D. Programme through the grant no. PHD/0193/2560. The funders had no role in study design, data collection and analysis, decision to publish, or preparation of the manuscript.

### Grant Disclosures

The following grant information was disclosed by the authors:
Mae Fah Luang University.
Royal Golden Jubilee Ph.D. Programme: PHD/0193/2560.

### Competing Interests

The authors declare there are no competing interests.

### Author Contributions

- Sarunpron Khruengsai conceived and designed the experiments, performed the experiments, analyzed the data, prepared figures and/or tables, authored or reviewed drafts of the paper, and approved the final draft.
- Patcharee Pripdeevech conceived and designed the experiments, analyzed the data, prepared figures and/or tables, authored or reviewed drafts of the paper, and approved the final draft.
- Chutima Tanapichatsakul performed the experiments, prepared figures and/or tables, and approved the final draft.
- Chanin Srisuwannapa performed the experiments, analyzed the data, authored or reviewed drafts of the paper, and approved the final draft.
- Priya Esilda D'Souza analyzed the data, authored or reviewed drafts of the paper, and approved the final draft.

- Parinya Panuwet conceived and designed the experiments, analyzed the data, authored or reviewed drafts of the paper, and approved the final draft.

## DNA Deposition

The following information was supplied regarding the deposition of DNA sequences:

The sequences are available at GenBank: MN704648, MW485486, and MW495050.

## Data Availability

The raw data are available in the Supplementary File.

## Supplemental Information

Supplemental information for this article can be found online at http://dx.doi.org/10.7717/peerj.11242#supplemental-information.

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
