# Peer review of "Antifungal properties of volatile organic compounds produced by Daldinia eschscholtzii MFLUCC 19-0493 isolated from Barleria prionitis leaves against Colletotrichum acutatum and its post-harvest infections on strawberry fruits"

_PeerJ, doi:10.7717/peerj.11242_

## Round 0.1 · original submission · Minor Revisions

Our reviewers enjoyed your work but have some suggestions for improvement, including some experimental work necessary to confirm the identification of the endophyte. Other suggestions of experimental work would, in my view, greatly improve your manuscript, but may not be strictly necessary for acceptance.

Reviewer 1 ·

Basic reporting

Self-contained with relevant results to hypotheses.

Experimental design

Original primary research within Aims and Scope of the journal.
Research question well defined, relevant & meaningful. It is stated how research fills an identified knowledge gap

Validity of the findings

All underlying data have been provided; they are robust, statistically sound, & controlled.

Additional comments

This article aimed to isolate endophytic fungi from Barleria prionitis plants grown in Thailand and to investigate the antifungal properties of the volatile compounds produced by these endophytic fungi against Colletotrichum acutatum.

The article is straightforward. Although the article is not innovative, it contains original and interesting information. The authors may consider removing the term “room temperature” throughout the manuscript. The term may misguide readers. Besides, this article would be improved if the authors clarify or revise the following:

Line 59. Italicize “C. acutatum”.
Line 104. Leave a space between “al.,” and “2008”.
Line 181. Revise to “… CD-AX, Japan)”
Line 273. Revise to “(HP 6890, Agilent Technologies)”.
Lines 350-351. Revise to “varying degrees (Fig. 1).” And remove “The results of the in vitro assay are shown in Fig. 1.”
Line 353. Revise to “(80.79%, p<0.05).”
Line 361. Revise to “incubation period (Figures not shown).”.
Line 398. Revise to “respectively (Figures not shown).”.
Line 403. Revise to “ treated with elemicin than the positive control sample”.
Line 428. Revise to “According to previous studies (Singh et al., 2005; Amoo et al., 2011), B. prionitis …”.
Lines 465-468. However, results shown in Figure 3 were not impressive. Elemicin demonstrated minimal effect against C. acutatum”.
Line 478. Revise to “… this plateau for additional 3 days.”

Table 1. The table should be self-explanatory. Inhibition, severity, and incidence percentages of what? observed from the in vivo experiments. Clarify how the authors obtained these data, referring to Figure 3. The strawberries treated with elemicin shown in Figure 3 seem to have less inhibition than 70.87%.

Figures 1, 2, 3, 4, and 6. Italicize fungus names in the titles.

Revise Figure 3 title to “In vivo antifungal activity of the volatile compounds produced by the Daldinia sp. MFLUCC19-0493 against C. acutatum infections on day 7 after incubation at 27°C.”

Reviewer 2 ·

Basic reporting

There were missing references that appeared in the text and are not in the reference list: (Sahu 1984) and (Guo et al 2018).
The discussion is too long because there is a lot of redundancy from the Results section.
Table 3 is unnecessary - the data is presented in Fig 4.
Figures 2 and 5 should be removed to supplemented materials.

Experimental design

The molecular identification of the endophytic fungus was performed using only one set of primers. The authors should validate their identification with at least 3 sets of primers. Therefore, the authors should sequence at least two more genes in order to confirm their results.

Validity of the findings

No comment

Additional comments

General comments:
Since the volatiles that were examined in this study are organic molecules, please change to VOCs (volatile organic compounds) instead of volatile compounds throughout the manuscript.
In the third paragraph of the introduction, the authors give examples of bioactive endophytic fungi. Since this study is about the endophytic fungus Daldinia, the authors should elaborate on this fungus and especially on its bioactivity described in the literature. See articles by Marc Stadler for example.
Molecular identification of the endophyte BP11 should be done with at least 3 different sets of primers. Therefore, the authors should complete the identification with 2 more sets of primers.
It is known that Daldinia produces volatiles with a pronounced, sweet and fruity odor [Stadler M, Læssøe T, Fournier J, Decock C, Schmieschek B, Tichy HV, et al. A polyphasic taxonomy of Daldinia (Xylariaceae). Stud Mycol. 2014;77: 1–143. pmid:24790283]. Did the author noticed a sweet smell from isolate BP11, and if so, please indicate it in the results, since this smell is also a feature of Daldinia.
From the screening of the bioactive endophytes, did the authors identified the second-best bioactive endophyte (BP24)?
The concentration units used in the manuscript are not clear. What do you mean by “µL/mL headspace”? Please clarify and change throughout the manuscript.
Did you check the viability of the pathogen in the in vitro experiment with the VOCs? This data is missing.
Did the authors examine the bioactivity of isolate BP11 against other phytopathogenic fungi (in in vitro assays)? In my opinion, a screen of the activity of this isolate against various phytopathogenic fungi in vitro may strengthen the importance of this isolate. For example, volatiles from D. cf. concentrica fully inhibited the growth of 12 phytopathogenic fungi (Use of the Endophytic Fungus Daldinia cf. concentrica and Its Volatiles as Bio-Control Agents
Liarzi O, Bar E, Lewinsohn E, Ezra D (2016) Use of the Endophytic Fungus Daldinia cf. concentrica and Its Volatiles as Bio-Control Agents. PLOS ONE 11(12): e0168242. https://doi.org/10.1371/journal.pone.0168242).
Did the authors try to prepare mixtures comprising different ratios of the most abundant VOCs? Perhaps a synthetic mixture might be more potent to protect the strawberry fruits from C. acutatum. If so, it will be easier to use the synthetic compound or mixture instead of the organism itself because it will not be necessary to maintain life-supporting conditions.
Can the author compare the concentration of elemicin used in the study and the concentration of this compound emitted by the BP11 isolate?
The discussion section is much too long and the parts describing the results of this study should be shortened.


Specific comments:
Line 42 - write "their volatile" insead of "the". Delete "by these endophtyic fungi"
Line 43 - delete "that usually causes" and write "the causal agent of"
Line 45 - write "a" instead of "the"
Line 46 - "in vitro" should be in Italic. Please correct throughout the manuscript.
Line 48 – write 80.3% - one number after the decimal point is enough. Please correct throughout the manuscript.
Line 48 - "in vivo" should be in Italic. Please correct throughout the manuscript.
Line 61 – the molecular identification of the BP11 endophyte should be described after the description that this endophyte possesses the highest activity – line 48.
Line 61 – instead of “based on the results of PCR reaction assay” write “molecular identification of BP11 endophyte revealed…”
Line 70 – change degrees to Celsius, since all the other temperatures in the manuscript are in Celsius.
Line 93 – delete “which are readily accessible”
Line 102 – please add reference to reinforce your statement.
Line 109 – I disagree with the statement that biofumigation does not require a sophisticated system. This is because you need maintain life-supporting conditions for the bioagent. The author should rephrase this sentence.
Line 119 – the reference for (Sahu, 1984) is missing.
Line 144 – didn’t the authors added antibiotic such as tetracycline or chloramphenicol to prevent bacterial growth during the isolation procedure?
Line 147 – delete “to prevent microbial contamination”.
Line 164 – it is not clear which primers did you use for the molecular identification of C. acutatum? Did you sequence the whole genome? Please clarify.
Line 188 – change to “viability” test
Lines 196 & 303 – delete the sentence. It belongs to the Results section and not to the Materials and Methods section.
Line 210 – “a 7-days old”
Line 210 – the endophyte culture plate in this test was without its cover. Please clarify.
Lines 224 & 299 – this is the same formula as in line 184. Therefore, please refer to the formula in line 184 and delete the formula in lines 224 & 299.
Line 231 – “as follows” and not “as followed”.
Line 238 – change “exposing to” instead of “treating with”.
Line 254 – why 30oC and not 27oC as the temperature used in the experiments?
Line 292 – please clarify how you calculated the estimated concentrations. You can calculate the moles of each compound used in the experiment (see for example Liarzi O, Bucki P, Braun Miyara S, Ezra D (2016) Bioactive Volatiles from an Endophytic Daldinia cf. concentrica Isolate Affect the Viability of the Plant Parasitic Nematode Meloidogyne javanica. PLOS ONE 11(12): e0168437.)
Line 305 – fashion? Perhaps manner is better word.
Line 315 – replace “above” instead of “previously”.
Line 348 – the reference to Fig. S1 should be moved to line 350, because the figure demonstrates only the bioactive isolates.
Line 350 – delete the word “but”.
Line 358 – this section should be combined with the previous paragraph and not as a separate section. Also, this section is written in a passive mode instead of active mode. Please rephrase.
Line 366 – please elaborate on each treatment. For example, the difference between the control and negative control treatments.
Line 368 – add (Fig. 3 C versus D) after “isolate BP11”
Lines 370-372 – delete, it is a repetition of the data presented in Table 1.
Lines 384-386 – write “The in vitro antifungal activity of the six most abundant VOCs against C. acutatum are summarized in Table 3.
Line 402 – change to Fig. 3E.
Line 405 – add “and similar to those obtained with the endophyte BP11”.
Line 413 – please clarify exactly what is each control.
Line 416 – please add “these results suggest that Both BP11 and elemicin are not phytotoxic to strawberry fruits”.
Lines 429-431 – please rephrase your hypothesis. This plant may serve as a candidate for isolation of endophytes that might be active against phytopathogens. I disagree that the bioactivity is only against C. acutatum.
Lines 432-444 – please refer to the suitable figures and tables in the text.
Lines 445-455 - compare the VOCs emitted by Daldinia isolate BP11 to published VOCs of Daldinia. For example: (Liarzi O, Bar E, Lewinsohn E, Ezra D (2016) Use of the Endophytic Fungus Daldinia cf. concentrica and Its Volatiles as Bio-Control Agents. PLOS ONE 11(12): e0168242. https://doi.org/10.1371/journal.pone.0168242) (Pažoutová S, Follert S, Bitzer J, Keck M, Surup F, Šrůtka P. et al. A new endophytic insect-associated Daldinia species, recognised from a comparison of secondary metabolite profiles and molecular phylogeny. Fung Divers. 2013;60: 107–123.)
Line 474 – the reference to Guo et al 2018 is missing.
Line 476 – why there is loss in the concentration of the synthetic VOCs? Possible reasons might be degradation of oxidation.
Line 515 – it is not clear if the farmers should use the endophyte itself or elemicin. Please clarify. The author should add disclaimers about the use of the fungus, for example, the need maintain life-supporting conditions.

Figs 2 & 5 – should be presented as supplemented materials.
Fig. 3 – write isolate BP11 instead of Daldinia sp. MFLUCC19-0493, since the results of the molecular identification is described only later in the manuscript. Also, the figure legend is not clear. Please elaborate and clarify.
Fig. 4 – the results demonstrated in the figure are redundant to Table 3. I would suggest deleting Table 3 or presenting it as a supplemented material.

---

## Round 0.2 · Minor Revisions

Please perform the additional sequencing with two more sets of primers required by our expert reviewer, to ensure absolutely unequivocal identification of the precise strain of your endophyte. You should also address the other minor points.

Reviewer 2 ·

Basic reporting

There is still a problem with the reference of Guo et al 2018. In the reference list, there is no such reference. Perhaps, the authors mean 'Gao et al 2018' and not 'Guo et al 2018', which was added to the revised manuscript.

Experimental design

The authors molecularly identified the phytopathogen C. acutatum using ITS primers. However, there is no indication in the text if indeed the pathogen was confirmed as C. acutatum (not % of identity, coverage, etc.).
Also, despite my previous suggestion to verify the molecular identification of the endophyte with two other sets of primes, the authors deposited only the ITS sequence of the endophyte in the GenBank database. The authors identified the endophyte using a phylogenetic tree. I would recommend verifying this identification by sequencing two more sets of primers and to deposit the resulting sequences in the GenBank, as the authors did for the ITS primers. This would strengthen the validity of the endophyte of interest.
Moreover, please add the accession numbers of the sequences used for the construction of the phylogenetic tree.

Validity of the findings

No comment

Additional comments

Line 114 - the organism 'Streptomyces globisporus' is a bacterium and not a fungus and therefore may not serve as an example for endophytic fungi.
Line 228 - replace 'in' to 'and'.
Line 359 - delete the '.' after (Fig. S1).
Line 361 - there is a space missing between '3.3' and 'and'.
Lines 373-374 - include in the parentheses the corresponding figures.
Line 468 - check the reference. Apparently, it should be 'Gao' and not 'Guo'.
Table 3 and Figure 2 - elaborate on the treatments: emphasis the differences between the controls.

I would also recommend deleting the first paragraph of the Discussion in order to shorten this section.

---

## Round 0.3 · Minor Revisions

Please take care of the final minor issues highlighted by our reviewer.

Reviewer 2 ·

Basic reporting

This is my second review of the manuscript. The authors fulfilled all of my comments and changed the manuscript accordingly.
However, there are some minor comments:
Line 172 - instead of "A gene region ITS" change to "The internal transcribed spacer (ITS)".
Line 172 - please add references to the ITS4 and ITS5 primer sequences.
Line 175 - replace "a" instead of "the".
Line 175 - write "performed" instead of "confirmed"
Line 176 - after the word "strawberries" please add "to confirm pathogen identification".
Line 329 - add "s" to the word "belong"
Line 427 - add "s" to the word "number"
Line 457 - there is a missing space between "19-0493" and "is"

Experimental design

The authors sequenced more genes for the molecular identification of the endophyte and also confirmed the pathogen using a pathogenicity test. In addition, the authors submitted the genetic sequences to the GenBank and listed their accession numbers. However, one of the accession numbers - MW495050 was not found in the Nucleotide database (due to today, 1st of February, 2021), whereas the two other accession numbers were valid. The authors should check why this accession number is missing and solve this issue.

Validity of the findings

No comment

Additional comments

Dear author
Thank you for revising your manuscript thoroughly. Yet, there is a missing accession number - the sequence for the RPB2 is not found in the Nucleotide database. The two other accession numbers - for the ITS and LSU sequences are valid and can be found in the Nucleotide database. Please check this issue. In addition, a few very minor changes in the manuscript are listed above. In my opinion, the manuscript is much better now and should be published once all accession number problem will be resolved.

---

## Round 0.4 · accepted · Accept

Thank you for addressing the last remaining issues!